# On AI-centered Retrosynthetic Planning: a Survey

## Abstract

Retrosynthetic planning is one of the most challenging problems in organic chemistry up to date. It involves the process of designing a target molecule by recursively decomposing it into simpler molecules through a series of backward chemical reaction steps. Typically, these simpler molecules are either commercially available or easy to synthesize. The development of new computer-aided retrosynthetic methods can advance the development of target molecules in drug design, material science and agrochemicals. In this paper, we present the retrosynthetic planning problem and the current AI-based methods. We conduct an in-depth review of the application of AI-based methods to retrosynthetic planning while also presenting the current challenges and potential future research directions.

## 1 Introduction

Retrosynthetic Planning (RP) is a branch of organic chemistry that was first formalized over four decades ago following the success of early backward chemical synthesis. It entails designing a chemical pathway for synthesizing a target molecule through a series of recursive backward reactions (Corey, 1967). Given a target molecule, the retrosynthetic plan decomposes it into simpler building molecules that are either commercially available or easily synthesizable. Subsequently, the building blocks can be chained up through a series of forward chemical reaction steps to synthesize the target molecule.

Retrosynthesis also applies to biochemistry in particular metabolic engineering (Lin et al., 2019). Bio-retrosynthesis differs from retrosynthetic planning in two aspects: (i) the reactions must be catalyzed by enzymes and, (ii) the building molecules are molecules that are naturally produced by the organism that is being used.

The success of RP can have potential benefits in the development of new drugs, vaccines, agrochemicals, materials science, target compounds etc. Development of new drugs for example is a long and expensive process and takes an average of 10-15 years and billions of money (DiMasi et al., 2016). In addition, the ability to design target compounds can accelerate the development of new chemical compounds which can help solve some of the most pressing challenges in the world today such as climate change and food security. Accordingly, the advances in computing power (Moore, 1998; Dean, 2020) and novel algorithmic designs e.g. Deep Learning (DL) (LeCun et al., 1998; 2015; Krizhevsky et al., 2012) have inspired a strong interest in the development of new computer-aided RP methods in the last few years.

Conventionally, the RP processes are heavily dependent on the expertise of organic chemists. In the last 5 years, however, computer-aided RP methods have gained more attention due to their ability to design better and faster planning routes (Segler et al., 2018; Chen et al., 2020; Han et al., 2022). Consequently, this has inspired the development of several molecular and chemical reaction databases e.g Reaxys[1], ChEMBL, (Gaulton et al., 2017), eMolecules [2], USPTO (Lowe, 2012) - to support such methods.

RP can be formulated as a tree. As an example, in Figure 1, $M_0$ is the target molecule. It can be designed by either the left or the right sub-tree. The squared internal nodes represent chemical reactions while the circled correspond to the molecules. Molecule $M_0$ can be synthesized by reaction $R_0$ or $R_1$. Reaction $R_0$ requires molecule $M_1, M_2$ and $M_3$ while reaction $R_1$ requires molecule $M_4$ and $M_5$. All the leaf nodes (encircled

---

[1]https://www.reaxys.com
[2]https://www.emolecules.com/

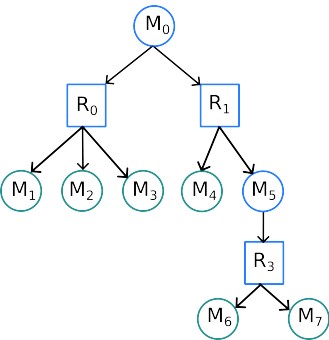

Figure 1: Retrosynthetic AND-OR tree plan. Squares represent reaction nodes and molecules represented by the circles. Each reaction node in the tree requires all of its children (AND node) while $M_0$ can be designed by the left or right sub-tree (OR nodes). All the leaf nodes are building block molecules.

green) are building block molecules e.g. commercially available or easy to synthesize. Molecule $M_5$ is not a building block thus it's decomposed into $M_6$ and $M_7$ through reaction $R_3$.

Based on the given AND-OR tree we can extract two retrosynthetic plans $P1$ and $P2$ for designing $M_0$ shown in Equations 1 and 2 where the symbol $\implies$ denotes the retrosynthetic symbol e.g. the backward chemical reaction. The route lengths of the plans $P1$ and $P1$ are 1 and 2 respectively. The route length is one of the commonly used metrics used to evaluate an effective retrosynthetic plan.

$$P1: \ M_0 \implies M_1 + M_2 + M_3 \tag{1}$$
$$P2: \ M_0 \implies \left\{M_6 + M_7\right\} + M_4 \tag{2}$$

In RP, the proposed routes must be effective, chemically viable and feasible. While shorter routes are generally preferred, this is however contingent on the availability of the building block molecules and the real-world costs such as human capital and labor. A transformation such as $M_0 \implies M_1 + M_2 + M_3$ is not unique. As a result, other transformations can be composed to design the target e.g $M_0 \implies M_a + M_b + ... + M_c$ given that $\left\{M_1, M_2, M_3\right\}$ and $\left\{M_a, M_b, M_c\right\} \in S$, where $S$ is the chemical space. While the planning length typically ranges between 15-20, the number of transformations at each level in the tree for simpler molecules ranges between 80-40000 (Segler et al., 2018) with complex molecules exceeding these ranges.

Currently, there are four key AI-based approaches to RP (i) expert-based systems, (ii) template-based (iii) template-free and semi-template. In **Expert-based**, RP is designed according to the expert-engineered rules where starting material is parsed with an inference engine and then matched against the Knowledge Base (KB) of chemical reactions (Pensak & Corey, 1977; Chen & Baldi, 2009). **Template-based** RP methods are based on the chemical reaction templates. Here, the reaction rules are not predefined as expert-based systems but rather extracted automatically from a database of chemical reactions. This is achieved by first generating the signature of the chemical reactions (reaction templates) associated with a given target molecule and then applying them to generate a list of potential reactants (Chen et al., 2020; Han et al., 2022; Kim et al., 2021; Segler et al., 2018). **Template-free** alleviates the template-based dependency by modeling RP as a prediction task. The product molecule is used as the input feature while the reactants are the labels. A learning algorithm is then used to learn the mapping between the input features and the labels (Liu et al., 2017; Karpov et al., 2019; Schwaller et al., 2019). **Semi-template-based** RP model RP as a two-step process involving (i) Product to Sython (P2S): the product molecule is broken into synthons(molecular segments) conforming with the reaction centers, and (ii) Sython to Reactant (S2R): the reactants for each synthon are then generated separately. Generally, the first step can be modeled as link prediction while the second uses several methods such as Seq2Seq, generative modeling with graphs, leaving group prediction, etc. (Somnath et al., 2021; Wang et al., 2020a; Shi et al., 2020). These methods are increasingly appealing

to the research community due to their ability to model diverse chemical relationships while also being interpretable.

Template-based methods are interpretable since they can be mapped to known chemical reactions in the database, however, they are limited by the chemical template database. Template-free methods have leveraged the advances in NLP e.g. attention (Bahdanau et al., 2014) and transformers (Vaswani et al., 2017) contributing to competitive performance in RP. While they are not as interpretable as template-based RP methods, they require no chemical template database. Template-free methods also require a database of chemical reactions in the form of product-reactant mappings. In addition, they are prone to generating invalid molecules due to their generative nature. Semi-template-based methods are interpretable and can model diverse chemical relationships while also being interpretable. While this class of RP design is increasingly appealing to the research community it inherits the limitations of molecular graph modeling as well as the sequence to sequence modeling as the second step is generative. Figure 2 summarizes the current AI-based RP methods.

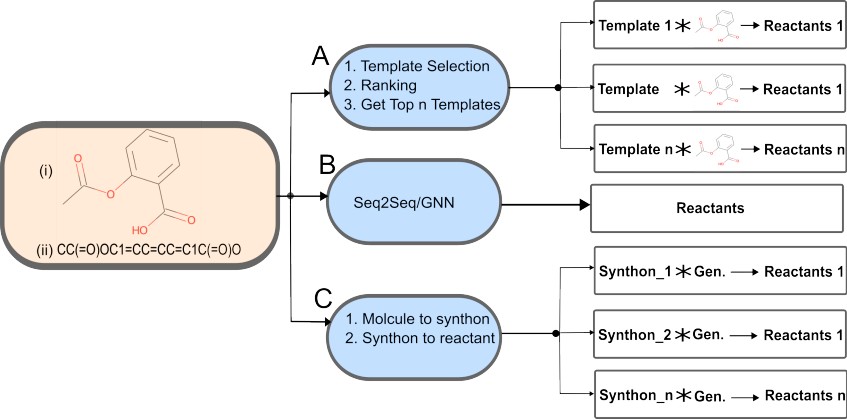

Figure 2: RP planning methods summary. A: template-based - generates a list of potential reactants by applying the predicted template, B: template-free - maps the product molecule to the reactants using Generative Models (Gen.) e.g. Seq2Seq, C: semi-template-based - breaks the product molecule into synthons and then generates the reactants for each synth. The input can be Molecular Graph (i) or SMILES (ii)- the two are equivalent.

## 2 Background

### 2.1 Retrosynthetic Planning Modeling

RP can be represented formally as a Markov Decision Process (MDP) $\mathcal{R} = \{S, A, P, R, \sigma\}$. The search space component $s \in S$ is a set of all chemicals in the organic chemistry with the terminal states corresponding to the building block molecules or molecules that cannot be validly decomposed further. An action $a \in A$ is a chemical reaction which transform the product molecule $s \in S$ into the reactant set $\{s'_1, s'_2, ...s'_n\} \in S$. The transition probability $P : S \times A \to 2^S$ is a one-step chemical reaction(one-step retrosynthesis) that accepts the product molecule $s$ and the action $a$ and then returns the reactant set. The reward function $R : S \times A \to \mathbb{R}$ is the cost single-step chemical reaction, while $\sigma$ includes a set of reaction parameters e.g. reagents that control the single-step chemical reaction. The mapping $P : S \times A \to 2^S$ is one-to-many mapping and in multi-step RP where there are several one-step retrosyntheses, the target product molecule $M_0 \in S$ is recursively decomposed into reactant molecules until the terminal state e.g. a given depth is reached or all building block molecules are found. Given that product-reactant pairs are not unique, the transition probability $P$ is non-deterministic and thus can result in exponential branching which is a major computational challenge in RP.

## 2.2 Definitions

**SMILES.** Simplified Molecular Input Line Entry System (SMILES) (Weininger, 1988) is a chemical coding standard that is used to represent a molecule as a string of characters. Each atom, bond, ring and other molecular properties are represented by a unique character. For example, SMILES for water ($H_2O$) is **O** while that of ethanol ($C_2H_6O$) is **CCO**. SMILES represent atoms by their symbols e.g. **C** for Carbon. Hydrogen atoms are not explicitly represented as they are inferred from the valency of the atoms. The standard also includes mechanisms of representing other molecular features such as bond types - single (**-**) or ( ), double (**=**) and triple (**#**), aromaticity, chirality and ring structures.

**SMARTS and Reaction Template.** SMILES arbitrary target specification (SMARTS) is also a chemical coding standard that is used to represent a subgraph pattern of a molecule. It is an extension of SMILES and every SMILES is a valid SMARTS. SMILES are mostly used to encode general molecular subgraph patterns which are key in chemical reactions, as such, they can be used to define the template of a chemical reaction. A reaction template is a subgraph pattern that is used to define a chemical reaction. A chemical reaction must satisfy the reaction template to be valid. SMARTS encoding follows the *reactant_pattern ≫ product_pattern* format where the *reactant_pattern* is the left-hand side of the reaction and the *product_pattern* the right-hand side. As a concrete example, **[#6:1][O:2]** ≫ **[#6:1]**=**[O:2]** is a SMART-based reaction template which takes any Carbon (atomic number 6 and the first atom in the reaction list) which is singly bonded to the Oxygen (second atom) and gives a product having double bonds (=). There are many reactions which can be derived from this template. Simply put, a reaction template defines subgraph patterns for a chemical reaction, thus, *a chemical reaction is an instance of a reaction template*. In template-based RP, reaction templates are defined as SMARTS.

**Molecular Fingerprints.** Molecular fingerprints are fixed-length binary bits that are used to represent a molecule. They are used to encode molecular features such as the presence of a particular atom, bond type, ring structure, chirality, substructure pattern, functional groups, unique ions, leaving groups, etc. Common fingerprint encoding methods include Extended-Connectivity Fingerprints (ECFP) (Rogers & Hahn, 2010) - topological fingerprint method encoding structural features - and Morgan Fingerprints (Rogers & Hahn, 2010) - a circular fingerprint method encoding key molecular properties by encoding the subgraph patterns of the molecule. Other fingerprint encoding methods are based on count vectors, hashing, etc.

**Molecular Graph.** A molecular graph is a graph representation of a molecule. It is composed of *atoms* and *bonds* which are represented by the nodes and edges respectively. The atoms are connected by the bonds. Formally, a molecular graph is a tuple $G = (V, E)$ where $V$ is the set of *atoms* and $E$ is the set of *bonds*. Each atom $v \in V$ is a tuple $v = (a, x)$ where $a$ is the atom type and $x$ is the atom feature vector. Each bond $e \in E$ is a tuple $e = (v_i, v_j, b)$ where $v_i$ and $v_j$ are the atoms connected by the bond $b$. GNN (Kipf & Welling, 2016) is a popular DL method which can be used to encode molecular graphs. GNNs are composed of several layers and each layer is composed of a message passing and a readout function. The message-passing function updates the node features while the readout function aggregates the node features to give the graph features. The graph features can then be used to predict the target molecule or other molecular properties.

**One-step and Multi-step Retrosynthesis.** A one-step retrosynthesis involves a single chemical reaction where given a target molecule, the objective is to design a valid set of reactants that can lead to the target molecule e.g. in Figure 1, $M_0$ is the target molecule while $\{M_1, M_2, M_3\}$ is the reactant set which obtained by through one-step retrosynthesis via reaction $R_0$. Simpler molecules can be designed through a single-step retrosynthesis, however, complex molecules require a series of one-step retrosyntheses: multi-step retrosynthesis i.e. in Figure 1, $M_0$ can be designed through the left or right sub-tree. The right sub-tree is composed of two one-step retrosyntheses (multi-step) as there are two reactions - $R_1$ and $R_3$ - which are required to design $M_0$. Most molecules require a series of one-step retrosyntheses to be designed.

**Synthon.** Given an atom-mapped product molecule and reactants, the changes in bonds and atoms can identified automatically between them. This is achieved by breaking the product molecule into segments called synthons. A synthon is a subgraph of the product molecule which is connected by the reaction center. In semi-template RP a molecule is first broken into synthons according to the identified reaction centers. Then, reactants for each synthon are generated separately and combined to give the target molecule (Shi et al., 2020).

## 3 Existing AI-based RP Methods

In this section, we present a general overview of existing AI-based methods while also presenting their taxonomy in Figure 3.

(i) **Expert-based:** These are the earliest RP methods designed according to the expert-engineered rules. RP research work under this category includes early models such as SYNLIMA (Johnson et al., 1989), WODCA (Gasteiger et al., 2000) and LHASA (Pensak & Corey, 1977). Generally, these methods code chemical reaction concepts such as leaving groups, functional groups, reaction centers etc. in a Knowledge Base (KB) and then attempting to match against the starting material provided by the user as a query. The models marked the shift from the manual to the computer-aided RP. Despite this, however, they are limited by scalability, generalization and the KB coding. This is despite the recent resurgence of modern rule-based methods such as Chen & Baldi (2009). The limitations and increase in computing power have inspired the design of new AI-based RP method: template-based, template-free and semi-template-based.

(ii) **Template-based:** Here, the reaction rules are not predefined as expert-based systems but rather extracted automatically from a database of chemical reactions. This is achieved by first generating the reaction template associated with a given target molecule and then apply them to generate a list of potential reactants. The template selection can be achieved by a learning model such as MLP neural network (Chen et al., 2020; Han et al., 2022) or by a template similarity measure (Coley et al., 2017). Currently, several template-based retrosynthetic models have been proposed leading to strong performance on several benchmark datasets. These include Retro* (Chen et al., 2020) an A*-like inspired tree search method with neural evaluation to select the most promising RP path. In Retro* the search tree is modelled as an AND-OR tree whereby each reaction node (OR node) can lead to a target molecule that requires all its children (AND nodes). Similar to Retro*, Kim et al. (2021) also uses an AND-OR tree to model the RP. However, in contrast to Retro*, their model learns to improve the policy by self-play - a concept from Reinforcement Learning (RL) where the model learns to improve its performance by playing against itself (Silver et al., 2017). Prior to the two RP models, (Segler et al., 2018) proposed a Monte Carlo Tree Search (MCTS) based model called 3N-MCTS which is a multi-step RP model that uses 3 deep neural networks (DNN) to model the expansion policy, filter policy and value function. The expansion policy is used to select the most promising templates and the filter policy is to select the most *realistic* reactions following after template application. The value network is used as an evaluation function during the tree search rollout phase. While 3N-MCTS recorded strong RP results it was limited by the tree representation scheme. Each node in the tree was modeled to include more than one molecule which proved more computationally expensive and limited their molecular design choice. This limitation was addressed by Kishimoto et al. (2019) which first showed that the AND-OR tree is an apt molecular representation scheme for RP. This design choice was adopted by (Chen et al., 2020; Kim et al., 2021; Han et al., 2022); Han et al. (2022) proposed a GNN-based model with route cost optimization. GNN-based models such as Han et al. (2022) have recorded strong performance due to their ability to model molecules as graphs which are more chemically aware compared to fingerprint methods e.g. Chen et al. (2020); Kim et al. (2021). Other notable template-based RP include template similarity ranking (Coley et al., 2017), template scoring and ranking based on the symbolic features (Segler & Waller, 2017), two-step reaction with reaction group prediction and molecule generation with DNN (Baylon et al., 2019), tree search with DL evaluation (Schreck et al., 2019), MCTS with self-playing policy improvement (Koch et al., 2019) and optimizing one-step RP with RL (Liu et al., 2023a). Overall, strong template-based RP methods are based on tree search with DL evaluation algorithmic design. While template-based methods are interpretable, they can only be mapped to known chemical reactions in the database thus limiting their generalization. This limitation has inspired the design of template-free methods.

(iii) **Template-free.** Here, the RP problem is modeled as a prediction task. The product molecule is used as the input feature while the reactants are the labels. A learning algorithm is then used to learn the mapping between the input features and the labels. Based on this, the RP problem can be designed as a classification task where the labels are the reactants. Alternatively, it can be designed

as a machine translation (a sequence-to-sequence) task where the product is the source language while the reactants are the target language. Many strong template-free RP methods are based on sequence-to-sequence models (Sutskever et al., 2014) e.g. LSTM Se2Seq (Schwaller et al., 2019), LSTM Seq2Seq with attention (Liu et al., 2017) and Seq2Seq with the transformer architecture (Karpov et al., 2019). While the transformer-based models have recorded state-of-the-art performance in the last five years due to the attention mechanisms (Bahdanau et al., 2014) to model inter-dependencies between the input tokens, they are limited by chemical awareness as SMILES input tokens are treated as characters and not chemical entities. This limitation has led to increased adoption of molecular graph-based RP modeling including graph translation model with edge attention (Xie et al., 2022), higher-order molecular graphical reconstruction (Jin et al., 2020) and end-to-end learning with reaction metadata (Coley et al., 2019b; Xie et al., 2022). Similar to template-based methods, GNN template-free methods have recorded better performance and are more interpretable compared to the sequence-to-sequence models. Despite this, template-free models are constrained by their generative nature and thus prone to generating invalid molecules: whether seq2seq or GNN-based.

(iv) **Semi-template.** Template-based methods are limited by the chemical template database. Template-free methods are limited by awareness and are prone to generating invalid molecules due to their generative nature. Semi-template RP attempts to some of these challenges by modeling RP as a two-stage process: (i) Product to Synthon (P2S) and (ii) Synthon to Reactant mapping. In the first stage, the product molecule is first broken into synthons according to identified reaction centers. This phase can be formulated as a link prediction problem (Daud et al., 2020; Rossi et al., 2021). In the second stage, the reactants for each synthon are then predicted separately and then combined to give the reactant set. This phase can be solved by deep generative graphs, Se2Seq or template-based methods. As an example, early semi-template-based work was presented by Shi et al. (2020) which proposed a product-synthon, synthon-reactant model. Product-synthon step was modeled as a link prediction between bonds and atoms using Relational GNNs (Schlichtkrull et al., 2018) while the synthon-reactant the step was modeled as a graph generation problem constrained as MDP. In Yan et al. (2020) the authors followed a similar design philosophy, however, the second stage was modeled using a transformer (Vaswani et al., 2017) based on SMILES. The last two laid the design pipeline for follow-up work: (i) Somnath et al. (2021); a product for reactant center prediction in the first stage is followed by leaving group prediction corresponding to each synthon segment, and (ii) Wang et al. (2020a); modeled both the first and second stages as machine translation while introducing novel methods for selecting potential reactants in the second phase.

## 4 AI-methods Overview.

A summary of template-based, template-free and semi-template methods is presented in Table 1. Key AI-based methods applied include (i) Tree Search, (ii) Proof Number Search (PNS), (iii) Machine translation (iv) Graph Neural Networks and (v) Reinforcement Learning.

**Tree search.** Tree search algorithms are effective in solving combinatorial optimization problems. A*-like methods (Liu et al., 2023a; Chen et al., 2020; Kim et al., 2021; Han et al., 2022) and Monte Carlo Tree Search (MCTS) (Koch et al., 2019; Segler et al., 2018) dominate template-based RP with state-of-the-art performance. A*-like methods are inspired by the A* search which is characteristically *optimal, complete and effective* including a cost function heuristic which helps to find paths leading to the goal faster.

MCTS incrementally builds a search tree while effectively balancing the exploitation and exploration (Kocsis & Szepesvári, 2006) in the search space. The algorithm is composed of four steps: (i) selection, (ii) expansion, (iii) simulation and (iv) backpropagation. In RP during the selection phase, the most promising reaction template is selected. The expansion phase involves applying the selected template to the product molecule to generate the reactant set. In the simulation phase, chemical templates are sampled from the policy function and then applied until the terminal state or given depth is reached. The backpropagation phase involves updating the Q values and the number of visits for each node in the tree with the reward e.g. +1 if the path leading to the terminal state includes valid building block molecules or -1 otherwise. Silver et al.

Table 1: Summary of the current AI-based RP methods.

| | Novelty | Multi-step | Fingerprints | SMILES/SMARTS | USTPO | Accuracy | Software |
|---|---|---|---|---|---|---|---|
| **TEMPLATE-BASED** | | | | | | | |
| Liu et al. (2023a) | One-step optmization with RL | x | x | x | x | 86-99 | Torch, RDChiral |
| Lee et al. (2023) | Rule generalization DL memory | x | - | - | - | 23-93 | RXNMapper |
| Han et al. (2022) | Reaction cost GNN optimization | x | x | x | x | 80-91 | Torch, RDChiral |
| Yu et al. (2022) | Goal driven Actor Criticv RL | x | - | - | x | 58-88 | - |
| Balzerani et al. (2022) | MCTS with DL & heuristics | x | - | - | - | 77-83 | RetroPath |
| Kim et al. (2021) | A*-like with DNN and self-play | x | x | x | x | 57-96 | Torch, RDChiral |
| Chen et al. (2020) | A*-like tree search with DNN | x | x | x | x | 79-86 | Torch, RDChiral |
| Wang et al. (2020b) | Self-play policy MCTS | x | - | - | - | 55-76 | ASKCOS |
| Kishimoto et al. (2019) | PNS with heuristic | x | x | x | x | 25-81 | TF1, RDKit |
| Koch et al. (2019) | MCTS with similarity ranking | x | - | - | - | 77-83 | RDKit, RetroPath |
| Baylon et al. (2019) | Template prediction with DHN | x | x | x | x | 58-82 | - |
| Schreck et al. (2019) | Self-play with DL evaluation | x | x | - | - | 64-73 | TF1, RDChiral |
| Dai et al. (2019) | GNN with sampling | x | x | x | x | 64-93 | Torch, RDKit |
| Delépine et al. (2018) | BFS template similarity ranking | x | x | x | - | 63-81 | RDKit |
| Segler et al. (2018) | MCTS with DNN Policy Rollout | x | x | x | x | 85-95 | - |
| Segler & Waller (2017) | Template ranking with rules | x | - | x | - | 62-99 | Keras, RDChiral |
| Coley et al. (2017) | Template similarity ranking | x | x | x | x | 52-92 | RDKit, RDChiral |
| **SEMI-TEMPLATE** | | | | | | | |
| Chen et al. (2023) | Sequential optimal S2R completion | - | - | x | x | 63-91 | Sklearn, Torch |
| Somnath et al. (2021) | P2S and leaving group predictions | - | - | x | x | 63-88 | - |
| Shi et al. (2020) | P2S & S2R w/ MG transformer | - | - | | x | 61-88 | Torch |
| Yan et al. (2020) | P2S & S2R w/ transformer | - | - | x | x | 62-80 | Torch, NMT |
| Wang et al. (2020a) | Both P2S & S2R as transformers | - | - | x | x | 64-86 | Torch, RDKit |
| **TEMPLATE-FREE** | | | | | | | |
| Liu et al. (2023b) | End-to-End In-context Learning | - | - | x | x | 60-94 | Torch, RDKit |
| Wan et al. (2022) | Transformer molecular attention | - | x | x | x | 57-99 | RDChiral |
| Irwin et al. (2022) | Transformer-based RP model | - | x | x | x | 53-95 | Deepspeed, Torch |
| Lin et al. (2022) | RP with Energy-based model | - | x | x | x | 52-94 | Torch, RDChiral |
| Tu & Coley (2022) | Local & global MG feature encoding | - | x | x | x | 52-85 | NMT, Torch |
| Xie et al. (2022) | GNN with edge attention | x | - | x | x | 50-72 | Torch, DGL |
| Sacha et al. (2021) | Graph Se2Seq w/ MG attention | - | - | x | x | 48-91 | Torch, RDKit |
| Sun et al. (2020) | Energy-based Seq2Seq with GNN | - | - | x | x | 53-85 | - |
| Schwaller et al. (2019) | Reactant-product attention | - | - | x | x | 76-92 | IBM RXN, NMT |
| Lin et al. (2020) | MCTS with transformer evaluation | x | - | x | x | 54-81 | TF1, RDKit |
| Coley et al. (2019b) | End-to-end Weisfeiler-Lehman GNN | - | - | - | x | 85-93 | TF1, RDKit |
| Zheng et al. (2019) | Self-correcting DNN with transformer | - | - | x | x | 74-81 | - |
| Karpov et al. (2019) | Template-free RP using transformers | - | - | x | x | 40-69 | TF1, NMT |
| Schwaller et al. (2018) | End-to-end seq2seq BiLSTM | - | - | x | x | 80-87 | RDKit |
| Bradshaw et al. (2018) | Electron Gated GNN | - | - | x | x | 70-94 | RDKit |
| Liu et al. (2017) | Product-reactant Seq2Seq | - | - | x | x | 37-70 | TF1 |
| Wei et al. (2016) | Fingerprint fusion with DNN | - | x | x | - | 85-86 | Skearn, Hyperplot |

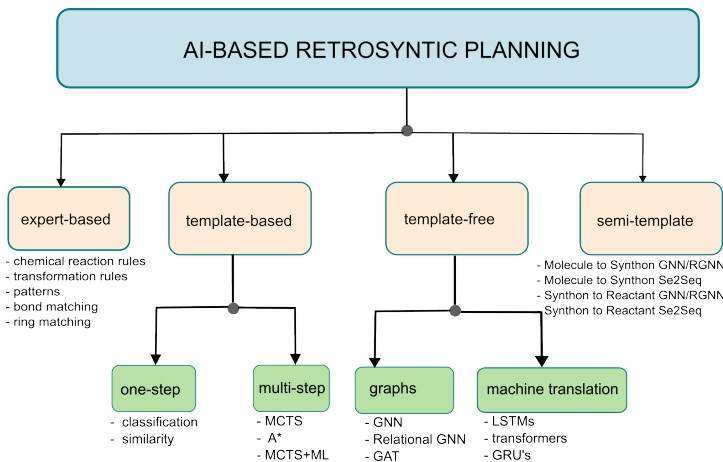

Figure 3: *Taxonomy of current of AI-based retrosynthesis.*

(2016) presented an effective mechanism of combining MCTS with DL as such the rollout phase of MCTS is increasingly being replaced with a prediction model. As reported, this design choice is influenced by the sub-optimality of the random policy. Similar designs have been employed by RP methods such as Koch et al. (2019). Current MCTS includes selection, expansion and backpropagation with DL as rollout policy. The pseudocode for this pipeline is presented in Algorithm 1.

---

**Algorithm 1:** MCTS with DL evaluation.

**1** $root \leftarrow \texttt{tree}(state)$;
**2 while** *time* **do**
**3** $\quad node \leftarrow \texttt{select\_node}(root)$;
**4** $\quad node \leftarrow \texttt{expand}(node)$;
**5** $\quad v \leftarrow \texttt{DL\_eval}(node)$;
**6** $\quad \texttt{back\_propagate}(node,\ v)$;
**7 return** $\texttt{best\_child}(root)$;

---

Apart from MCTS and A*-like methods, other tree search algorithms that have been applied to RP include Breadth-First Search (BFS) and Depth First Search (DFS). BFS is a tree traversal algorithm that searches for all the nodes *level-by-level*. It does not require any heuristic or value function to determine the node visit but rather *uniformly*. In contrast, DFS visits all the nodes by going as deep as possible and then backtracks if the search value is not found. DFS and BFS methods have recorded very promising results in Delépine et al. (2018) while also playing a key role as baseline models Han et al. (2022); Kim et al. (2021); Chen et al. (2020); Segler et al. (2018).

**Proof Number Search (PNS).** PNS can be used to model the game state in a perfect information game using an AND-OR tree structure. During the playouts, the maximizing player is mapped to the AND tree nodes and the minimizing player to the OR nodes. For each node in the game tree, the proof and disproof numbers are recorded and subsequently updated. During the expansion, nodes with the least proof value and nodes with the least disproof values are selected for OR and AND nodes respectively. RP can be represented as an AND-OR tree such that given the target molecule, it can be synthesized by either the reaction node (OR) and each reaction node requires all its children (AND). Many sequential and parallel variants exist including PPNS, PN2 and DFPNS. Modeling RP with an AND-OR tree is sound and effective, thus, recent methods are increasingly employing this modeling approach (Kishimoto et al., 2019; Kim et al., 2021; Chen et al., 2020) recording state-of-the-art results. These implementations are reinforced with DL evaluation functions or policies.

**Machine translation**. These algorithms are based on the encoder-decoder architecture where the input sequence is encoded into a fixed-length vector by the encoder and then decoded into the output sequence by the decoder (Sutskever et al., 2014) (Seq2Seq). The Se2Seq models were proposed to solve the machine translation problem in NLP. However, through transfer learning, they have been increasingly adopted to solve other hard NLP problems. Further, following the introduction of the attention mechanism (Bahdanau et al., 2014; Vaswani et al., 2017), Seq2Seq models have gained more traction due to their ability to model long-range dependencies leading to better performance across many NLP tasks. Seq2Seq is a primary algorithmic design in template-free RP. Existing RP methods are based on three key approaches including Seq2Seq LSTM(Schwaller et al., 2019), Seq2Seq with attention (Lin et al., 2020) and Seq2Seq with transformer (Karpov et al., 2019).

**Graph Neural Networks (GNNs).** This is a class of neural networks suited to learn and generalize on graph-structured data (Kipf & Welling, 2016). Molecules are inherently graphical and as such GNNs are increasingly being used to model them. While the input feature in RP can be fingerprints, SMILES or molecular graphs, the latter is more chemically aware and as such is gaining more attention in all categories of RP methods e.g. template-based (Han et al., 2022), template-free (Xie et al., 2022) and semi-template (Shi et al., 2020).

**Reinforcement Learning (RL).** RL algorithms learn by interacting with the environment. The agent learns policy or value function which maximizes the reward by taking actions in the environment (Sutton & Barto, 2018). While not very common in RP, possibly due to convergence and reward design issues, some studies have explored this approach. Liu et al. (2023a) proposed an RL-based algorithm composed of two policy network networks, a synthesizability value network and a cost value network. The policy networks are used in the planning phase while the value networks in the update phase to evaluate the most promising targets. Koch et al. (2019) on the other hand proposed an RL-based algorithm similar to AlphaZero (Silver et al., 2017) which combines MCTS with DL while improving the policy through self-play. A closely related work by Yu et al. (2022) employed a goal-driven actor-critic RL algorithm to solve the RP by learning the policy and value function.

Table 1 shows a summary of the current template-based and template-free methods. The abbreviations in Table are as follows; The Novelty summarizes the key idea in the paper, Multi-step: whether the method is multi-step or one-step, Fingerprint: whether the method uses fingerprints as input features, SMILES/SMARTS: whether the method uses SMILES or SMARTS as input features, USTPO: whether the method uses the USPTO dataset or its variant, Accuracy: this corresponds to the performance ranges of the method according to Top @K accuracy where the value of K ranges across different methods, Software: the software used to implement the method. The abbreviations used in the table are as follows: TF1: TensorFlow V1, GNN: Graph Neural Network, DNN: Deep Neural Network, USTPO: US Patent Office Dataset (Lowe, 2012), RDChiral (Coley et al., 2019a), DHN:Deep Highway Network Srivastava et al. (2015), ZINC (Irwin et al., 2012), MG: Molecular Graph, P2S: Product to Synthon, S2R: Synthon to Reactant, RXNMapper (Schwaller et al., 2021), AGREDA (Blasco et al., 2021), RetroPath: RetroPath RL: (Delépine et al., 2018), NMT: OpenNMT (Klein et al., 2018), deepspeed: (Aminabadi et al., 2022),

On the methodology, template-free methods are limited by their generative nature however the ability to include chemical awareness and modeling global and local molecular-based attention such as in Wan et al. (2022) have shown that their performance can be improved. Semi-template methods are increasingly gaining attention due to their diverse ability to include chemical awareness in the design process. Since their inception by Shi et al. (2020), they have increasingly recorded better performance up to date. Somnath et al. (2021) and (Shi et al., 2020) even reported instances where such methods recorded better performance than template-based. Generally, template-based are more stronger and interpretable. At the feature level, while fingerprint encoding has dominated template-based methods, other molecular representations such as InChI (Heller et al., 2015) have also been used - though not very common as only a single work has reported it (Koch et al., 2019). Top@K accuracy is the primary evaluation metric used by the majority of current RP methods. It involves computing the percentage of the target molecules that are included in the top K predictions. About the datasets, the majority of the template-based methods used the USPTO (Lowe, 2012) database. Template-free methods, on the other hand, used a subset of USPTO (USPTO-5OK) or custom datasets. This is similar to semi-template methods. The USPTO is composed of more than 1.8M chemical

reactions from 1976 to 2016. Other databases have also been used by some methods including DrugBank (Wishart et al., 2008), LASER (Wishart et al., 2008), Retrorules (Duigou et al., 2019) and literary texts (Delépine et al., 2018). Major building block molecule databases include ZINC (Irwin et al., 2012), eMolecules [3], and Reaxys [4],

## 5 Open challenges

(i) **Metrics.** There are no widely agreed metrics for evaluating the performance of AI-based RP methods. While shorter routes are generally preferred, other factors such as the ease of synthesis from the proposed plans or the availability of reagents to catalyze the reactions are also key. Based on this constraint, some longer planning routes can be preferred over the shorter ones. Also, in template-free retrosynthesis, different scoring metrics have been proposed by different methods e.g. Pearson coefficient and top@K accuracy. These evaluation variations currently, present a challenge while conducting an in-depth comparative analysis.

(ii) **End-to-end design with metadata**. Reagents are chemical compounds which catalyze chemical reactions. The majority of the existing template-based and template-free solvers do not include the reagents in their design pipeline. Given that the reagents contain rich metadata information, their inclusion in the the design process has two key benefits (i) soundness: every chemical reaction is composed of reactants, *reagents* and products, (ii) robust DL-based evaluation models: DL-based evaluation networks can be trained on the richer dataset.

(iii) **Databases**. USPTO is the major publicly available reaction database for both template-based and template-free methods. Currently, over 95% of the existing methods are trained and tested on this dataset. The creation of new databases that can allow for true *out of sample* validation and testing is thus imperative. Importantly, such new datasets can also include a uniform evaluation metric for both the template-based and template-free methods. In consequence, this is likely to help address metric variation inconsistencies as previously discussed. Additionally, other researchers have proposed retrosynthetic solvers based on custom datasets e.g. extraction from the literary text. While this approach is a very promising research frontier, it is likely to compound the comparative analysis issues.

(iv) **Sparsity.** Molecular fingerprints encode the presence of subgraph features in a molecule as a sparse row vector. Such, among others may include the presence of functional groups, rings or bond types. The existing template-based and template-free methods are dominated sparse molecular fingerprints method - ECFPF and Morgan Fingerprints- as such they inherit the sparsity issues such as the inability to capture *richer* information and increased computational complexity for longer fingerprints.

(v) **Natural product synthesis.** Synthesis of naturally occurring products though existing RP methods are still a challenge. This is attributed to the fact that the majority of the existing RP methods are limited by the chemical reaction databases. This limitation has been reported in a number of RP studies including Lin et al. (2020) and Segler et al. (2018). In contrast, in bio-retrosynthesis, several successful natural product syntheses have been reported (Lin et al., 2019; Zheng et al., 2022).

## 6 Future Research Directions

In this section, we present a summary of the key future research directions. Firstly, the **template adaptation.** The current template-based methods use DL greedily selects the most promising templates which are consequently applied to a target molecule to generate the reactant set. While this approach has recorded strong and very promising results, designing RP methods that not only select the templates to apply but also adapt the *template to the molecule* or *template to the route length* can lead to better planners. These adaptations can allow the RP solver to learn reusable templates that can be applied to future similar states in the design process.

---

[3]https://www.emolecules.com/
[4]https://www.elsevier.com/solutions/reaxys

Secondly, representation of **Molecules as Graphs** is a very promising research frontier as molecules are inherently graphical. In addition, graph-based representations such as GNNs are more chemically aware and thus can capture richer features compared to molecular fingerprints. In addition, the design choice can also help address the sparsity issues associated with molecular fingerprints. Thirdly, **Stronger RL Policies**, since Silver et al. (2017) the random simulation phase has been replaced with a prediction model as a result of the sub-optimality, in consequence of the sub-optimality. While some existing template-based RP methods have increasingly adopted these new design choices there is a need to design models with stronger policies. Fourthly, **On generative Modelling**, the Se2Seq has inspired the development of strong template-free methods such as (Lin et al., 2020). While the transformer Seq2Seq model is currently phasing out the LSTMs, the current limitation is still chemical awareness. The transformer can model the long-range dependencies in the input sequence, however, it is still limited by the SMILES-based character-level attention. While the model proposed by Wan et al. (2022) is a step in the right direction towards modeling Seq2Seq with molecular attention, it is still limited by expressiveness.

Fifthly, **Active and transfer learning** is a key design choice. Molecules share a lot of chemical similarities. The classical reaction USPTO database, for instance, contains over 1.8M reaction samples. Many of these samples have similar rings, bonds or functional group patterns which once learned, can be *transferred* or *generalized* to help learn faster and reduce the search space. These properties can be achieved by transfer learning or active learning methods, or by a combination of the two (active transfer learning).

Lastly, **End-to-end RP, parallelization and multi-task learning** are performant design approaches in many machine learning problems as such can also be applied to RP. End-to-end learning with products, reagents, reactants and catalysts in the design pipeline is key to designing robust RP methods. The majority of the existing methods, however, do not explicitly model the reagents and catalysts RP components. The majority of the existing methods are not parallelized by design. Designing parallel RP methods has two main advantages (i) improved performance: the workload is distributed and (ii) richer representations: each parallel instance can learn local representations which can be *shared* globally. The current RP methods are embarrassingly parallel, however, *less than 1%* are parallelized by design. Multi-task learning is ubiquitous in modeling computational biology problems. They are generally characterized by *the ability to optimize more than one objective*. In template-based RP for instance a template filter network is trained separately from the template selection policy yet the two can be learned by optimizing a joint objective function. Similarly, template-molecule and template-route adaptation can also be learned in an end-to-end fashion instead of separating the two. Such models are easier to train and allow for the *sharing* of the learned representations leading to more robust models.

## 7 Conclusion

In this work, we presented a comprehensive survey of the current AI-based retrosynthesis methods: expert systems, template-based, template-free and semi-template. We presented a taxonomy of the current method according to their design philosophy. We also presented a summary of the key AI-based methods applied in retrosynthesis. We then presented the current open challenges and future research directions. We hope that this work will help researchers understand the current state of the art and also help in the design of stronger retrosynthetic planners.

## Ethical Statement

We have no ethical issues to report about this research work.

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
