# OpenReview forum: "On AI-centered Retrosynthetic Planning: a Survey"
_TMLR — Rejected by TMLR_

### Review · Reviewer_BmP5 · 2024-02-06

**Summary Of Contributions:**

This is a review/survey paper of AI methods for synthesis planning, covering different reaction prediction models and search algorithms.

**Audience:**

No

**Claims And Evidence:**

No

**Requested Changes:**

TMLR's acceptance criteria are 1) well-supported claims 2) interest to the audience. It is not clear how a review paper fits into these criteria (even if it were well-written) since review papers often do not make claims or results (unless they do something like a meta-analysis). Therefore, even if the errors and mistakes mentioned in the previous section of my review are fixed I am not sure that this paper should be accepted. Broadly, to be accepted I think this paper needs to provide some novel insights to the reader, but it is not my job as the reviewer to come up with these insights myself.

**Strengths And Weaknesses:**

Broadly I think the paper mentioned almost all the relevant literature that it should have, although a notable exception was generative models for synthesis plans [1-2]. Other than that, I had a negative impression of the paper. My main criticisms are:

1. Disagree with conceptual organization of the paper
1. Lots of mistakes and subtle errors
1. Too many vague/empty statements and superficial descriptions of papers
1. Questionable value over other review papers on the same topic

I will elaborate on these points below.

## 1) Conceptual organization

Three things stand out to me the most.

First, the categorization into expert-based, template-based, template-free, and semi-template-based seems misleading to me, especially because "expert-based" systems are not mutually exclusive with the others. My understanding is that expert-based systems are more or less template-based systems where the templates are chosen by experts (and typically assigned in a hand-coded way rather than by a machine learning model). Methods like [3] which combine automatic and manual template curation also show the shortcomings of this paper's taxonomy.

Second, multi-step planning methods are discussed under template-based methods when they are in fact agnostic. This is a *huge* error, since a primary intended use case of many template-free and semi-template-based methods is in multi-step search. Recent works have even deployed this successfully [4-5]. I think this presentation fundamentally mis-represents this line of work.

Finally, the layout of the paper is a bit odd. Notably, section 3 and section 4 have a lot of overlapping content, e.g. the discussion of multi-step planning and "machine translation"/"template-free" methods. It reads as if two authors wrote sections independently and then combined them.

## 2) Mistakes and errors

A huge number of statements in the paper were either wrong or "not quite right". There are too many to list exhaustively so I will just focus on one big ones and one small one from every page.

Big mistake: in section 2.1 the authors state that retrosynthesis can be formalized as an MDP whose state space is the set of all molecules. However, they define the transition probability as mapping a state-action pair to a *set* of states, when in an MDP it should in fact map to a *single* state (or more generally a distribution over states). The authors then vaguely state how mapping to a set of states means that each state needs to be solved recursively without really explaining this. I am somewhat sympathetic to the authors because I acknowledge retrosynthesis can *almost* be formulated as an MDP, but details matter and retrosynthesis is in fact *not* an MDP: this is why previous works use an AND/OR tree. It can however be converted to an MDP if the state space is defined as sets of molecules: this is what Segler et al did.

Here is one small mistake from every page:

1. The authors state that algorithms like MCTS and retro* inspired the development of datasets like USPTO and ChemBL when it was very obviously the other way around (datasets enabled development of algorithms). I have no idea why the authors would think this: the datasets were published *before* the algorithms and referenced in these papers...
2. The authors try to write a general reaction $M_0\Rightarrow M_a + M_b + \ldots + M_c$, but having only defined $M_a,M_b,M_c$ this statement this does not specify a general reaction with $N$ reactants.
3. The MDP error mentioned above
4. They emphasize "a chemical reaction is an instance of a reaction template". While not technically false, multiple templates can also produce the same reaction, so there is a "many-to-many" relationship between reactions and templates which this sentence does not describe precisely.
5. Template-selection MLP is attributed to Chen et al 2020 and Han et al 2022 instead of e.g. Segler et al 2017.
6. A* search is defined as being "characteristically optimal, complete and effective". However, this depends on the search heuristic being admissible. Most search heuristics will not have this property.
7. Putting multi-step methods in the template-based block of the table
8. "Current MCTS includes [...] DL as rollout policy". Many current methods, e.g. AiZynthfinder, still use rollouts.
9. Molecular graphs are described as "more chemically aware" than SMILES when in fact SMILES uniquely define a graph, so they contain equivalent information.
10. "Reagents are chemical compounds which catalyze chemical reactions." -> regents *cause* chemical reactions. They can be catalysts or reactants.
11. Again, confusing reagents and catalysts.

## 3) Empty/vague statements

Scientific writing should be precise and informative. However, the overwhelming majority of the analysis in this paper is superficial, supported by very vague statements without citations. Some random examples:

- "The existing template-based and template-free methods are dominated sparse molecular fingerprints method - ECFPF and Morgan Fingerprints- as such they inherit the sparsity issues such as the inability to capture richer information and increased computational complexity for longer fingerprints" (page 10). What sparsity issues? What is the richer information that is not captured?
- "Template-free alleviates the template-based dependency by modeling RP as a prediction task." (page 2) All these methods use prediction tasks. How is this one different?
- "Despite this, template-free models are constrained by their generative nature": what is the constraint from their "generative nature"? Is the problem not more lack of constraints?

More broadly, the description of many papers feels like a 1-sentence summary that I would get from ChatGPT. I think the paper is missing unifying themes and deeper analysis of these works.

## 4) Other reviews

There are other reviews on this topic which contain very similar content, e.g. [6-7]. It is not clear what value this paper brings over these existing reviews.

## References


[1] Barking up the right tree: an approach to search over molecule synthesis DAGs

[2] Learning To Navigate The Synthetically Accessible Chemical Space Using Reinforcement Learning

[3] Influence of Template Size, Canonicalization, and Exclusivity for Retrosynthesis and Reaction Prediction Applications (10.1021/acs.jcim.1c01192)

[4] Re-evaluating Retrosynthesis Algorithms with Syntheseus

[5] Models Matter: The Impact of Single-Step Retrosynthesis on Synthesis Planning

[6] Recent advances in artificial intelligence for retrosynthesis (10.48550/arXiv.2301.05864)

[7] Artificial Intelligence for Retrosynthesis Prediction (10.1016/j.eng.2022.04.021)

---

### Review · Reviewer_PNLL · 2024-02-07

**Summary Of Contributions:**

This paper reviews existing works on reaction prediction and retrosynthetic planning. It explains the underlying concepts and tries to systematically categorize existing approaches on several axes. Finally, it provides an outlook for what future work in this domain may bring.

**Audience:**

Yes

**Broader Impact Concerns:**

As this is a review paper, I believe it poses no ethical risks or concerns.

**Claims And Evidence:**

Yes

**Requested Changes:**

While the work done by the authors is a good initial step, I believe the paper would need a major revision to be acceptable for publication at TMLR. I would request for the points listed in the "Weaknesses" section be addressed first.

**Strengths And Weaknesses:**

=== Strengths ===

(S1): Retrosynthesis is an important topic in computer-assisted drug design. Summarizing the existing body of works is a useful and timely effort, especially given the generally growing interest in applying AI to scientific domains.

(S2): The work does a reasonable job gathering many references to prior works, also trying to cover older expert-based approaches.



=== Weaknesses ===

(W1): The paper makes fundamental errors confounding single-step reaction prediction with multi-step planning. In most cases, single-step prediction (predicting viable reactant sets given a product) is considered a separate problem from multi-step planning (chaining single-step predictions into entire synthesis plans), and the techniques are usually different as well (the former is a prediction problem, while the latter is a planning problem). There are exceptions to this that e.g. feed an entire search graph into an ML model [1], and it's important to benchmark both elements jointly [2, 5], but in most cases the choice of the single-step model and the choice of the search algorithm are nearly orthogonal. In contrast, the proposed paper confuses the two, and claims several times (both in the text, e.g. "strong template-based methods are based on tree search", and in the taxonomy in Figure 3) that multi-step search algorithms are a subclass of template-based models, which is just not true; many prior works run search using template-free or semi-template-based models as well [2-4]. I would recommend making a clear distinction between single-step and multi-step, noting where the algorithms can be combined freely and where there could be potential problems or constraints. On that note, I don't like the design of Table 1 that lists papers and contains a binary "Multi-step" column to denote whether a particular paper has done something with multi-step, because it again confounds the fact that some papers introduce both a single-step model and a planning algorithm, but the former is not "multi-step", it just happens to appear in a paper that also does multi-step benchmarking. I would rather see two separate tables for single-step and multi-step, and also perhaps a separate note of which pairs were so far combined with each other in existing works.

(W2): Overall I found that the way prior works are organized and summarized is not very helpful. For example, the main taxonomy in Table 1 includes a binary "USPTO" column for whether a particular paper used one of the USPTO derivatives. This is not inherent to a particular method, and so I'd rather not have that in the high-level comparison, and additionally have separate tables to aggregate results per dataset, differentiating between different USPTO derivative datasets appropriately. It also has an accuracy column which is said to contain some top-k values but the k and dataset varies; the paper thus shows statistics of the form "paper A reported accuracies between 58 and 88, while paper B reported between 64 and 73", but these were measured differently on different datasets, so it doesn't seem particularly useful. I would hope for a more systematic and useful breakdown of the different methods, which would require many tables looking at different axes of comparison.

(W3): I found the paper rather hard to parse, and I feel that a reader who is not already familiar with the discussed concepts would not benefit from reading it in the current form, or may even become confused due to unclear explanations. Some examples of statements which wouldn't be clear to a reader who is new to the topic:

- "RP can be formulated as a tree." - again we should be clear we mean multi-step planning here, and "tree" should probably rather say "tree search problem"

- "number of transformations at each level in the tree for simpler molecules ranges between 80-4000" - unclear what this refers to (number of valid single-step reactions for a single target molecule? number of nodes at a particular depth in a typical search?)

- "Template-free methods also require a database of chemical reactions in the form of product-reactant mappings" - unclear if this refers to mappings in the sense of atom mappings (which, by the way, are referred to at the bottom of page 4 yet don't seem to be explained), or mapping in the sense of mapping products to valid sets of reactants; in the former sense this sentence is not true, and in the latter sense it seems obvious, as it's essentially stating that a template-free model also requires training data (which is true for all models)

- "σ includes a set of reaction parameters e.g. reagents that control the single-step chemical reaction" - not clear how this feeds into the MDP (one could imagine this to be included in the output from a reaction prediction model and thus part of the transition P?)

- "mapping P : S × A → 2^S is one-to-many mapping" - unclear (my understanding is that P is a mathematical function and thus returns a single reactant set; perhaps if one imagines a mapping from S without A then it would be one-to-many)

- "transition probability P is non-deterministic and thus can result in exponential branching" - unclear

- Discussion of parallelization is not clear (e.g. unclear whether the parallelization is across multiple target molecules or across expanding a search tree for a single molecule in several places at the same time)

(W4): While many existing papers and references are already included, a review paper should hopefully be exhaustive, so some more references listed in the "References" section of my review could also be included. Specifically, things that could also be mentioned include:

- Pistachio dataset [8] which was also used in some of the works, either for testing generalization [2] or for pretraining [10]

- More single-step models: [7]

- More search algorithms: [1, 9]

- Papers performing a holistic evaluation of several approaches and discussing potential pitfalls: [2-5]

Finally, there is also some work on retrosynthesis-driven drug-design [6] which often doesn't follow the backward retrosynthetic search blueprint, which may be out of scope for this review, depending on the authors' preference.

(W5): The paper appears to be generally not very well-written, containing typos throughout (see "Nitpicks" section for some examples).



=== Nitpicks ===

Below I list various grammar issues, typos, or broken sentences; this list is not exhaustive but is a representative sample of where one could start improving the work.

- "involves the process of designing a target molecule by recursively decomposing it into simpler molecules" (abstract) - the word "designing" could be confusing here, as one could imagine it refers to drug-design, whereas typically retrosynthetic planning is done with a concrete target molecule in mind (although there are exceptions of course [6])

- "Squares represent reaction nodes and molecules represented by the circles."

- misusing "e.g." in places  "i.e." would fit better

- "are interpretable and can model diverse chemical relationships while also being interpretable"

- some punctuation marks missing e.g. after "g" in "e.g."

- Figures 2 and 3 largely overlap in their purpose

- "molcule" in Figure 2

- "SMILES are mostly used to encode general molecular subgraph patterns" - do you mean SMARTS?

- "SMART-based"

- "e = (v_i ,v_j ,b) where v_i and v_j are the atoms connected by the bond b" - explain b

- "obtained by through"

- "can identified automatically"

- "bond types - single (-) or ( )" - please explicitly explain that single bonds are sometimes implicit instead of writing "( )"

- "Semi-template RP attempts to some of these challenges"

- "Se2Seq"

- "Criticv"

- "Retrosyntic" in Figure 3

- "deepspeeed"

- "optimizing one-step RP with RL"

- "as a result of the sub-optimality, in consequence of the sub-optimality"



=== References ===

[1] Liu et al, "FusionRetro: Molecule Representation Fusion via In-Context Learning for Retrosynthetic Planning"

[2] Maziarz et al, "Re-evaluating Retrosynthesis Algorithms with Syntheseus"

[3] Hassen et al, "Mind the Retrosynthesis Gap: Bridging the divide between Single-step and Multi-step Retrosynthesis Prediction"

[4] Torren-Peraire et al, "Models Matter: The Impact of Single-Step Retrosynthesis on Synthesis Planning"

[5] Tripp et al, "Re-evaluating Chemical Synthesis Planning Algorithms"

[6] Bradshaw et al, "A Model to Search for Synthesizable Molecules"

[7] Zhong et al, "Root-aligned SMILES: A Tight Representation for Chemical Reaction Prediction"

[8] Mayfield et al, "Pistachio: Search and Faceting of Large Reaction Databases"

[9] Tripp et al, "Retro-fallback: retrosynthetic planning in an uncertain world"

[10] Jiang et al, "Learning chemical rules of retrosynthesis with pre-training"

---

### Review · Reviewer_SWGn · 2024-02-10

**Summary Of Contributions:**

This paper surveys the literature on the topic of  AI-based retrosynthesic planning (RP), and categorizes methods in four groups: expert-based, template-based, template-free, and semi-template methods. A brief introduction to chemical synthesis, retrosynthesis, and the relevant concepts within that space is provided. A number of papers are surveyed and categorized, and their primary contribution is identified. An overview of training methods and paradigms is then presented, also categorized into tree search, proof number search, machine translation, graph, and reinforcement learning methods. Finally, some open challenges and future directions are identified.

**Audience:**

No

**Broader Impact Concerns:**

AI & ML methods in and surrounding material and chemical sciences may have profound impact on the world, including through potential misuse. This should at least be acknowledged in the paper. Please refer to the [TMLR Ethics Guidelines](https://www.jmlr.org/tmlr/ethics.html)

**Claims And Evidence:**

No

**Requested Changes:**

- "Sython" should be "Synthon"

I'm not sure I'm fully on board with the MDP formulation proposed in this paper:
- There is a mismatch between the transition function $P: S \times A \to 2^S$ and the choice of state space $S$. Transition functions in the original MDP framework are $S \times A \times S \to [0, 1]$ maps. Here the notation and the text suggest instead that we are using a map to all possible subsets of $S$ (which, yes in some sense is the case, because you may need multiple reactants to perform a reaction);
- "Given that product-reactant pairs are not unique, the transition probability P is non-deterministic"; this seems like an odd way to model things, some retrosyntheses have multiple reactants, so instead of acknowledging that as an integral part of the problem definition, it feels like this detail is (wrongly) translated to being distributional. What is the distribution?
- Many of the papers referenced in Table 1 do not fit this MDP, nor in fact fit into the MDP framework. This seems like a misuse of the MDP framework. In general, my understanding of these works is that they solve a particular kind of search problem whereby methods yield *subtrees* of "the retrosynthetic graph" (e.g. what is shown in Fig 1), not trajectories within that graph (which is what the MDP framework is almost systematically used for)

A better approach in my humble opinion, if the authors truly want to formalize this as an MDP, is to have states be either sets of molecules, or partial synthetic routes, but I don't think this matches much of the literature explicitly; different papers have different underlying search structure (which they often do not specify, unfortunately). In specifically RL works that do use an MDP, e.g. Schreck et al. or Koch et al., the former trick (states are sets) is used (albeit implicitly).

Alternatively, the authors could forego providing a formalism. Formalisms in a survey paper can be useful if they unify/generalize from many papers, but they are not necessary.


Section 3 is kind of just a wall of text. Readability would highly benefit from this being broken down further into paragraphs or even subsections rather than it being just a list. More generally, there are many places where adding more whitespace, more paragraphs, idea separations, would be beneficial. There are no space limits in TMLR.

Section 3:
- it would be good to explain more specifically what differentiates Expert-based systems from templates and e.g. priors/heuristics on retrosynthetic policies. Are KBs essentially structured templates? If not, how so?
- I don't understand this sentence: "This [what does This refer to? Creating templates? Applying templates?] is achieved by first generating [what does it mean to generate a template?] the reaction template associated with a given target molecule [target for what? where does this molecule come from?] and then apply[ing] them [what is them? the templates?] to generate a list of potential reactants"
- On template-based methods, such papers usually do one or both of two things: (1) come up with a way to generate a list of templates/SMARTS from some chemical database, (2) come up with a way to use templates to perform retrosynthesis. It would be good to properly separate and survey these two aspects of template-based methods.
- "machine translation (a sequence-to-sequence)", this doesn't seem like the right terminology, Seq2Seq models are _way_ more general than translation, they just learn a mapping $A\to B$ where $A$ and $B$ happen to be sets of sequences rather than pixels, vectors or tensors. I'm not sure why this analogy is used; I know it was used _historically_ as a metaphor to help readers since at the time virtually all Seq2Seq models were used for MT, but we know better now.
- templates come from chemical databases (which, isn't really expanded upon unfortunately), what kind of training data do template-free and semi-template methods use? How is it created?
- What is "Se2Seq"? (I'm assuming a typo? but it is repeated many times)

Section 4
- I'm not sure I see the point of Algorithm 1. It's basically just MCTS where it's pointed out that the heuristic comes from a deep model rather than an expert-written heuristic. Similarly to the MDP formalism, this feels like having an Algorithm just for the sake of having an Algorithm. Also, MCTS (and the set of tree search algorithms mentioned) is something that as far as I know is taught in typical CS undergrad curricula and does not need to be reintroduced here.
- I'm not sure what this sentence means: "Seq2Seq is a primary algorithmic design in template-free RP", "algorithmic" as opposed to what? Or do you mean that Seq2Seq models are mostly just used in template-free RP?
- GNNs and Seq2Seq are a model choice, not a method. I believe a missing category in this section is simply "Maximum Likelihood", which is how Schwaller et al. and following works (and I believe all template-free methods) train their Seq2Seq models. The particular choice of sequence pairs whose likelihood is being maximized is quite important (e.g. describing what's in USPTO_* and how it's processed would be good). The authors may also find it relevant to add a section in the paper that describe the various existing choices of AI models and architectures, and what their trade offs are to represent molecules and related data.
- This sentence isn't clear to me: "[Top@K] involves computing the percentage of the target molecules that are included in the top K predictions", which predictions though? There are all sorts of methods in this paper, with models that predict all kinds of things, values, expansion policies, token sequences, entire synthetic routes. Is this about local steps? the number of valid retrosynthetic routes in the top K?

In Section 5
- I'm not sure I see how _sparsity_ is an open challenge. Fingerprints have some well-understood drawbacks, but the open challenge is much more general than that--in my opinion, the challenge is simply "how should we [learn to] represent molecules?". More and more works are using deep learning in one way or another, often at parity with or exceeding FP-based baselines. Are the authors arguing for more research on FPs specifically?
- on Natural product synthesis, this naively feels like it should be part of datasets being a limiting factor. Why do Natural products deserve their own section as an open problem? They're certainly a very important application of retrosynthesis, but do they have unique _technical_ problems that no other applications of RP have? This survey seems to be more about methods than exactly what their applications are, so this paragraph seems out of place.

In Section 6
- I don't know what it means to adapt a template to a molecule, is this something that has been proposed in prior work?
- What does it mean for a policy to be stronger? Why is that an important factor in RL/RL for RP?
- "as a result of the sub-optimality, in consequence of the sub-optimality" (is the repetition an editing typo?) the sub-optimality of what? Do you mean that the predicted value in AlphaZero & co is not a monotone heuristic?
- on transfer learning, I'm not sure what is being transferred. Is the suggestion to pretrain models on vast quantities of chemical data and to then finetune them on USPTO?
- "current RP methods are embarrassingly parallel, however, less than 1% are parallelized by design" This seems like a contradictory statement, do you mean that, in theory, 99% of papers _could_ be implemented in a parallel way but that in practice papers' code use 1 CPU and/or 1 GPU? This is an engineering challenge, not an AI challenge (not to diminish the work of engineers, but it a separate type of expertise than AI/ML science, I'm not sure why it belongs in this paper). Also, where does that 1% number come from, is it an actual account of surveyed papers (there are fewer than 100 citations, so this seems unlikely) or just a figure of speech? If it is the latter, I would recommend not using a number, perhaps just a quantifier such as "very few".
- on multi-task learning, what is an example of multiple tasks here? Multiple possible sets of building blocks or reactions?


There are occasional typos and awkward turns of phrases in the paper. I would recommend further proofreading. Be mindful of the use of `\citep` vs `\citet`.

I hope that my comments will be helpful to the authors. Survey papers are great and helpful educational material for both newcomers and veterans desiring a refresher, and there are certainly a lot of things going on in retrosynthesis. I do encourage the authors to keep working on this paper.

**Strengths And Weaknesses:**

The paper surveys the field, providing some bases for readers new to AI-based RP.
Except for the MDP formalism which I have some gripes with (see below), I don't think there's any wrong or misleading claim in this paper.
The paper offers to identify salient challenges, and some promising or new and underexplored avenues.

The paper doesn't provide an especially deep historical account of the field, nor a significant landing pad for newcomers to either chemistry or machine learning. While the survey part covers the field fairly well, the paper falls flat in terms of providing deep insights and motivating open problems. In many cases, the authors do not elaborate enough so that a reader unfamiliar with the problems at hand would (a) understand why these problems matter nor (b) understand where to start addressing these problems.

Finally, and this is probably the main weakness of the paper, the presentation could be improved a lot. The order and organization of ideas, the way they are cut up into sections, the many unclear statements, all make the paper harder to read and its helpfulness as a survey paper greatly diminished. I elaborate on many such points below.

---

### Decision · Action_Editor_fkSR · 2024-03-12

**Recommendation:** Reject

**Comment:**

All reviewers unanimously voted to reject the paper. They appreciated the focus on retrosynthesis and organizing a large number of relevant works. However, due to the issues with the clarity of writing and the accuracy of different statements, the paper at the moment is not ready for publication. Based on that, I must recommend rejection at this stage. Thank you for your submission. I hope the comments will prove helpful in improving your work.

**Audience:**

The paper would find its audience due to the growing interest in AI for retrosynthesis. However, the clarity of writing significantly limits its accessibility.

**Claims And Evidence:**

Reviewers pointed out significant issues related to the accuracy with which the survey presented the field, notably and most concretely:
1. Confounding single-step and multi-step retrosynthetic analysis,
2. Issues with MDP formalization

**Resubmission Of Major Revision:**

The authors may consider submitting a major revision at a later time.